# Conditioning Factors of Linearized Wood’s Function Lactation Curve Shape Parameters, Milk Yield, Fat and Protein Content in Murciano-Granadina Primiparous Does

**DOI:** 10.3390/ani10112115

**Published:** 2020-11-15

**Authors:** Juan Vicente Delgado Bermejo, Francisco Antonio Limón Pérez, Francisco Javier Navas González, Jose Manuel León Jurado, Javier Fernández Álvarez, Luis Telo da Gama

**Affiliations:** 1Department of Genetics, Faculty of Veterinary Sciences, University of Córdoba, 14071 Córdoba, Spain; id1debej@uco.es (J.V.D.B.); fco.antonio.lp@gmail.com (F.A.L.P.); 2Centro Agropecuario Provincial de Córdoba, Diputación Provincial de Córdoba, 14071 Córdoba, Spain; jomalejur@yahoo.es; 3National Association of Breeders of Murciano-Granadina Goat Breed, Fuente Vaqueros, 18340 Granada, Spain; j.fernandez@caprigran.com; 4CIISA—Centre for Interdisciplinary Research in Animal Health, Faculty of Veterinary Medicine, University of Lisbon, 1300-477 Lisbon, Portugal; ltgama@fmv.ulisboa.pt

**Keywords:** dairy goat, peak production, persistence of lactation, Wood’s linearized model

## Abstract

**Simple Summary:**

Strategies aiming to improve milk yield and quality are essential to maximizing profitability and dairy goat chain efficiency, which in turn optimizes the commercialization of the products derived. A total of 137,927 official controls were traced and recorded from 22,932 Murciano-Granadina primiparous goats between 1996 and 2016. In this regard, the effects of conditioning factors related to kidding (such as type, year and season) and farm were evaluated to detect the phenotypic sources of variation of milk yield, fat content, protein content and curve shape parameters in Murciano-Granadina primiparous goats to provide useful information for dairy goat early selection.

**Abstract:**

A total of 137,927 controls of 22,932 Murciano-Granadina first lactation goats (measured between 1996–2016) were evaluated to determine the influence of the number of kids, season, year and farm on total milk yield, daily milk yield, lactation length, total production of fat and protein and percentages of fat and protein. All factors analyzed had a significant effect on the variables studied, except for the influence of the number of kids on the percentages of fat and protein, where the variation was very small. Goats with two offspring produced nearly 15% more milk, fat and protein per lactation compared to goats with simple kids. Kiddings occurring in summer–autumn resulted in average milk, fat and protein yields nearly 14, 19 and 23% higher when compared to winter–spring kiddings. Lactation curves were evaluated to determine the effects of the number of kids and season, using the linearized version of the model of Wood in random regression analyses. Peak Yield increased by about 0.3 kg per additional offspring at kidding, but persistence was higher in goats with single offspring. The kidding season significantly influenced the lactation curve shape. Hence summer-kidding goats were more productive, and peak occurred earlier, while a higher persistence was observed in goats kidding during autumn.

## 1. Introduction

The Murciano-Granadina Goat is the most important Spanish dairy goat, both in the number of individuals and production levels. There are more than 500,000 animals, of which 104,010 registered to the official herd book. They are spread across Spain, especially in Andalusia, Murcia and Castilla la Mancha, but an increasing number of animals are now reared abroad, mainly in other European countries, North Africa, Latin America and the United States [1].

Murciano-Granadina dairy biotype [2] is eumetric and characterizes by a subrectilinear or rectilinear profile. The coat is uniformly black or mahogany (stains of any color are unacceptable), with black or pink mucous membranes, respectively. Males’ average height and weight are 77 cm and 65 kg, while females’ average height and weight are 70 cm and 50 kg, respectively. Horns are not normally present, although they can appear in certain animals. Additionally, its continuous polyestrous nature makes the breed enjoy great prolificacy and longevity (6 births) [3,4].

Murciano-Granadina breeds’ average milk production is 584.4 kg, with an average lactation length of 287 days. Average milk fat and protein contents are 5.3% and 3.6%, respectively [1,4]. Despite its traditional semi-intensive handling, a tendency to intensification has progressively arisen until recent years. This newly implemented system uses close-to-farm natural resources and byproducts generated by agriculture.

Its important social-cultural and environmental roles are key for the maintenance and expansion of the population in rural areas. In these regards, the breed benefits from resources that cannot be used by other species. For instance, Murciano-Granadina goats feed on pastures from uncultivated areas and agricultural byproducts. Its great adaptability makes it especially suitable for cleaning arid or semi-arid scrubland zones, with its direct repercussions in fire prevention [5].

Quantitative and qualitative milk production are multifactorially affected. These factors can be grouped into animal-intrinsic (breed, size and body weight, age at parity and lactation number, litter size, physiological state, etc.) and extrinsic factors (Feeding, breeding system, kidding season, milking, dry period length, among others) [6].

The study of the lactation curve traces the evolution of populational or individual variations occurring in milk production through lactation. It enables detecting unexpected deviations at herd level as consequences of a deficient or erroneous diet or underlying pathologies in herds. According to Wood [7], knowledge of the lactation curve is necessary to determine the nutritional and reproductive management of lactating animals. This can be done by estimating the total production per lactation and curve shape parameters (lactation peak and persistence).

Conditioning factors not only influence milk production but also affect curve shape. Generally, the lactation peak is reached between the fourth and seventh weeks, subsequently decreasing afterward until the end of lactation. First-lactation goats usually present average lower productions and shorter lactation lengths when compared to goats at their second lactation or over.

The lactation curve can be described through several mathematical functions [3]. Such functions normally comprise an ascending phase, a production peak and a descending phase. By fitting these mathematical functions, the data obtained from different milk recording schemes can be used to predict milk yield and trace the evolution of total production. One of the most efficient functions, thus most frequently dealt with, is the equation proposed by Wood in 1967 [8].

The objective of this paper was to evaluate the effects of the number of kids, season, year and farm on total milk yield, daily milk yield, lactation length, total fat and protein production and percentages in primiparous goats. Additionally, lactation curve shape parameters (peak and persistence) were estimated and evaluated to determine whether these may be conditioned by the number of kids and season, using the linearized version of the model of Wood in random regression analyses.

## 2. Materials and Methods

### 2.1. Animal Sample and Production Records

First-kidding goats’ production records were provided by the National Association of Breeders of Murciano-Granadina Caprine Breed. The complete database comprised a number of 2,834,425 controls collected between 1996 and 2016 at 184 farms. Production controls were performed using the A4 method, following the recommendations of the International Committee for the Control of Animal Performance (ICAR) [9]. Out of the complete database, only those animals with standardized lactations lasting <300 days, at least 3 controls, with around 30 days intervals between controls, and up to a maximum of 5 offspring/parturition were considered in the study sample. Controls between which time intervals were longer than 70 days were discarded.

After editing, a total of 137,927 controls was available, corresponding to 22,932 lactations (Table 1) comprised the study sample and was used for statistical analyses. Approximately 67% of parities were simple, and only about 1% were triple (Table 2). Around 71% of parities took place during autumn and winter. The highest number of deliveries in Murciano-Granadina occurred between 2005 and 2013. However, a significant reduction in the number of parities has recently occurred. Parity number/farm ranged from 1 to 751, with an average of 124 ± 146 parities/farm (Appendix A).

### 2.2. Variables and Factors

The variables studied were total milk production (kg), average daily production (kg/day), lactation length (days), protein and fat total production (kg) and protein and fat percentages (%). The fixed effects (factors) studied were farms (*n* = 184 farms), parity year (*n* = 20, from 1996 to 2016), number of kids (1, 2, ≥3) and parity season (winter, spring, summer and autumn).

### 2.3. Total Production Calculation

Total milk production, fat and protein calculation were performed using the method by Fleischmann [9]. The estimate of total milk production (MP) is obtained by interpolation in the lactation curve, as expressed below:(1)$$MP=d_{1}MP_{1}+\left(d_{2}-d_{1}\right)*\frac{P_{1}-P_{2}}{2}+\left(d_{3}-d_{2}\right)*\frac{P_{2}-P_{3}}{2}+\ldots +d_{n-2}-d_{n-1}*\frac{P_{n-1}-P_{n-2}}{2}\;+d_{n}-d_{n-1}P_{n-1}$$

In which *P*_1_, *P*_2_, …, *Pn* are the kilograms of milk (or amount of fat or protein) produced in 24 h of the control day and *d*_1_, *d*_2_–*d*_1_, … *d*_n_–*d*_n−1_ are the intervals, in days, between the delivery and the first control (d1) and between successive controls (*d*_1_ and following), respectively.

Fleischmann method computes total production by adding the cumulative production in different sections, defined by the controls carried out. Thus, the days from birth to the first control are multiplied by the production in the first control to obtain the cumulative production in the first section. In the following controls, the average of successive controls is obtained [for example, (*P*_1_ + *P*_2_)/2] and multiplied by the interval between controls (*d*_2_–*d*_1_) to calculate the cumulative production between controls 1 and 2. Afterward, this is repeated for the remaining controls. At the end of lactation, production is computed after multiplying the production in the last control by the interval from the last control to drying.

### 2.4. Milk Fat and Protein Content Determination

To calculate the total production of fat and protein in lactation, fat and protein percentages in each control are multiplied by the amount of milk in the same control and standardized using the Fleischmann methodology.

### 2.5. Statistical Analyses

#### 2.5.1. Factor Effect Quantification

The total amount of milk, fat and protein produced per goat and lactation, as well as the average daily milk production, lactation length and fat and protein percentages, were subjected to an analysis of variance (ANOVA) with the PROC GLM of SAS to detect differences in the least-square means across farms, parity years, number of kids and parity seasons.

#### 2.5.2. Curve Shape Parameter Evaluation

The validated data for milk production, fat and protein contents derived from the different production controls were analyzed with the PROC GLIMMIX of SAS to build lactation curves. In this analysis, the parameters of the curve were considered as regression coefficients with a random and normal distribution, without any particular structure (type = un). The curves were estimated individually for each animal (subject = animal). The fixed effects of the number of kids (1, 2, ≥3) and parity season (winter, spring, summer, autumn) were considered. To fit the lactation curve, the (1967) was used. The incomplete gamma function of Wood can be expressed using the following formula:[MP_(t)_ = a t^b^ e^(−ct)^](2)
where MP (t) is the production of milk on day t of lactation; a is the parameter related to the level of production at the beginning of lactation; b and c are the growth and decrease rates of milk production, and e is the basis of natural or Napierian logarithms.

To solve the previous equation, the PROC GLIMMIX routine of SAS was transformed into its logarithmic form as follows:ln(y) = log(a) − ct + b Log(t)(3)

The aforementioned estimated parameters were used to calculate the production at lactation peak (PP) and the day on which lactation peak (DP) took place:PP = a (b/c)^b^ e^−b^(4)
DP = b/c(5)

Additionally, persistence percentage (P%) was estimated as the production in a given month, compared to previous months and calculated for the time intervals (t) = 50–100, 100–150 and 150–200. Then, persistence was expressed as P% 100; P% 150 and P% 200, respectively.

## 3. Results

### 3.1. Descriptive Statistics

Table 3 shows the level of significance, the least-square means, the coefficient of determination (R^2^) and the residual standard deviation (RSD) of each of the different variables studied across the levels considered in the study.

The results for average milk production were 301.03 ± 125.74 kg, with an average milk daily production of 1.44 ± 0.46 kg/day and a lactation length of 206.1 ± 24.7 days. The average fat and protein production percentages were 3.55 ± 0.3% and 5.08 ± 0.7% for protein and fat, respectively. This translated into a total of 10.94 ± 4.45 kg of protein and 15.57 ± 6.4 kg of fat per lactation. The different variables studied closely resemble the normal distribution (*p* < 0.05); hence a parametric approach was considered.

### 3.2. Effect of Number of Kids

Table 4 shows the results for the least-square means for the different variables studied across the number of kids possibilities (single, twins and triplets). Average total milk production difference of 40 kg was found between singleton and twin-pregnant goats and of 38 kg between twin and triplet-pregnant goats, respectively. Average daily production was 1.34, 1.51 and 1.62 kg/day, and average lactation length was 195.9, 199.8 and 206.5 days for singleton, twin and triplet-pregnant goats, respectively. Fat percentage was 5.17% for singleton and twin-pregnant goats and 5.19% for goats pregnant with triplets or more kids, which corresponded to a total fat production of 13.64 kg, 15.79 kg and 17.96 kg, respectively. Protein percentage was 3.56%, 3.55% and 3.53% for singleton, twin and triplet-pregnant goats, respectively. These values corresponded with a total protein production of 9.43 kg, 10.85 kg and 12.20 kg, respectively. The reduced variation in the percentages of fat and protein across the number of kids resulted in a lack of significant differences in the least-square means across the number of kids possibilities, as shown in Table 3 (*p* > 0.01).

### 3.3. Effect of Parity Season

Table 5 reports the least-square means values for the variables tested in the study across parity seasons (winter, spring, summer and autumn). Total average productions per lactation of 287.2 kg, 278.5 kg, 329.9 kg and 318 kg were reported in winter, spring, summer and autumn. The maximum difference of 51.4 kg was found between summer and spring. Summer was the season for which maximum productions were reached, while the lowest total milk productions were found in spring. Average daily productions of 1.48, 1.45, 1.50 and 1.52 kg/day were reported for winter, spring, summer and autumn, respectively. Average lactation lengths were 192.4, 188.7, 214.4 and 207.2 days during winter, spring, summer and autumn, respectively.

On one hand, total fat production and fat percentage for winter were 14.38 kg and 4.97%, respectively. The values of 14.08 kg and 4.98%, 17.96 kg and 5.41% and 16.875 kg and 5.35% were found for spring, summer and autumn, respectively. On the other hand, total protein contents and percentages of 10.04 kg and 3.45%, 9.74 kg and 3.45%, 12.08 kg and 3.66% and 11.45 kg and 3.62% for winter, spring, summer and autumn were reported, respectively.

### 3.4. Effect of Parity Year

Milk production varied across the years, following cyclic oscillations of about 80 kg between the least-square means of the years for which minimum and maximum production was reported (Figure 1). Maximum average productions ranging between 300 and 350 kg were found for the years 1996, 2001, 2006, 2011, 2015. The percentage of fat described an upward trend from 1999 (4.83% on average), reporting increasing percentages until 2011 (average of 5.33%), which stabilized thereafter. Protein percentage described oscillations through the years. The highest percentages of around 3.68% were found during the last years, during which the study took place. Total fat and protein production and average daily milk production reported similar oscillating patterns to those reported for milk production.

### 3.5. Effect of Farms

Average milk production ranged from 220 to 440 kg, while fat and protein percentages ranged from 4.5 to 6.5% and from 3.3 to 4.1%, respectively. As a result, average fat and protein production per lactating goat varied between approximately 9 and 30 kg and 6 and 21 kg, respectively. Frequency histograms for the variables studied across farms are reported in Figure 2.

### 3.6. Lactation Curve Shape Parameters

#### 3.6.1. Effect of Type of Kidding

Table 6 reports a summary of the average curve shape parameters across the number of kids. The graphical representation of the lactation curves for each number of kids possibility is shown in Figure 3. The linearized Wood function was fitted to the milk records with the parameters associated with initial daily yield (a), rate of increase prior to the peak yield (b) and rate of decrease after it (c). Figure 3 shows the lowest initial production is reached from goats pregnant with singletons (a = 0.65 kg), with this being slightly surpassed by twin-pregnant goats (a = 0.75 kg). Higher initial production was reported for goats pregnant with triplets or more kids (a = 1.1 kg).

Regarding the production in the peak (PP) and the day when the peak is reached (DP), it is observed that in goats with three or more kids, the highest productions are obtained in the peak (PP = 1.86 kg), and the peak is reached earlier (DP = 45), if we compare it with goats with one kid (1.35 kg at 57 d) and with goats with two kids (1.59 kg at 58 d), indicating that the average productions in the peak increase near 0.25 kg for each additional kid.

On the other hand, if we go to study the descending phase of the curve, we observe that the slope is less marked and with greater persistence in goats with one kid and two kids, there being few differences between both, in such a way that the persistence at 100, 150 and 200 days, respectively were about 96, 89 and 87% in goats of 1 and 2 kids and of 92, 88 and 86% in goats of 3 or more kids (Table 6).

#### 3.6.2. Effect of Kidding Season

Table 7 shows the data referring to the lactation curve according to the season of kidding. These results are represented in Figure 4. It was observed as the season when the average initial productions are lower is in winter (a = 0.48 kg), followed by autumn (a = 0.73 kg), spring (a = 0.86 kg) and finally the summer (a = 1.36 kg), where the highest initial productions were reached, being a difference between winter and summer of 0.88 kg of milk. The milk production peak of 1.51 kg was reached on day 23 in summer, much earlier than in other seasons. It was followed by spring, for which the highest production peak of 1.47 kg occurred on day 40. Milk production peaks of 1.44 and 1.41 kg were reached on day 68 for autumn and on the 61st day of lactation for winter.

Regarding persistence values, parities taking place in autumn presented higher persistence values (99, 93 and 91% at 100, 150 and 200 days of lactation) while parities taking place in spring reported the shortest persistence values (90, 85 and 83% for the same aforementioned stages of lactation). Curves from goats kidding in summer started with average higher yields and reached a production peak shortly afterward. Then, the descending phase is characterized by a smooth slope, which translated into a constantly maintained relatively high production.

## 4. Discussion

Average production values in first lactation goats in the present study were lower than the values reported in other studies. For instance, León et al. [10] reported productions of 372.58 kg for the first lactation of goats of the same breed. However, the findings in this study were similar to those by other authors. In these regards, Martínez Navalón and Peris Ribera [11] conducted a study between 1999 and 2002 and reported average production of 309 kg for first-parity goats with lactations of 216 days on average. Similar values were also reported by Pérez [12] between 2002 and 2012. Our results are in line with those reported by the Union of Farm Associations for Milk Control in Castilla y León [13] from controls first- natural lactation goats collected from 2015, 2016 and 2017, reporting average productions of 425.65, 401.10 and 453.60 kg, respectively in lactations with an average length of 254.47, 248.16 and 255.02 days, respectively.

Differences between Murciano-Granadina goats milk production and that from other Spanish native breeds have been reported in the literature. For instance, Florida breed primiparous goats milk productions of 412.83 kg in 225 days lactations have been reported [14]. However, García et al. [15] reported higher average yields of 441.07 kg and longer average lactation lengths (276.74 days) for the Florida breed. Malagueña primiparous goats presented comparatively higher average milk productions of 380.9 kg of lactations with average lengths of 256 days [16]. Contrastingly, lower milk productions of 286.85 kg with lactations of 210 days have been reported for the Palmera dairy goat [17]. Among all the Spanish breeds, Payoya production values were the closest ones to those in our study, with values of 314.2 kg and average lactation lengths of 234 days [18].

The values for fat and protein percentages reached the upper margin of the results found in the bibliography for Murciano-Granadina goats. Hence, similar results were found for Murciano-Granadina goats officially controlled in Castilla León. Fat and protein percentages were 4.98 and 3.56% in 2017, 5.33 and 3.55% in 2016 and 4.94 and 3.66% in 2015. Pérez [12] obtained similar results as well for their 10-year-long study involving farms of CAPRIGRAN Y ACRIMUR, results which were also similar to those reported by Martínez Navalón and Peris Ribera [11] who reported a protein percentage of 3.55% and a fat percentage of 4.76% in the same breed.

Compared to first birth goats from other native breeds such as Florida, Malagueña or Palmera, fairly similar fat and protein percentages were found, with small variations occurring in regards to fat percentage. For instance, in Florida, goat fat percentages of 4.9% and protein percentages of 3.54% were found [14]. In Malagueña goats, lower fat percentages with 4.27% and higher protein percentages of 3.64% were reported [16], while the primiparous Palmera breed goats presented similar fat percentages (5.04%) to those in the present study and higher than 4.44% protein percentages for standardized lactations at 210 days [17].

Compared to the productions reported in the literature for other highly productive European breeds, great differences can be found. For example, in controls carried out in the Alpine breed caprine breed in 2016 in France, much higher milk productions were found for primiparous goats, with 856 kg of milk in lactations with 308 days on average. The milk from these goats also presented lower fat and protein percentages (3.89 and 3.35%, respectively). In the Saanen breed in France, first-kidding goats reached productions of 1000 kg of milk, in lactations with an average length of 337 days and with fat and protein percentages of 3.69 and 3.23%, respectively [19].

Prolificity has been reported to be responsible for remarkable alterations in milk yield and composition. In fact, the tendency to present higher numbers of kids may be the result of selection with adequate reproductive management. In the present paper, higher dairy productions for multiparous goats were found, which was also found in publications by other authors such as Hayden et al. [20]; Salvador and Martínez [21]; Martí Vicent [22]; Pérez [12], which suggests litter size has a positive correlation with milk production. This may be due to a higher volume of the placenta in animals with more than one kid, which determines a higher production of placental lactogen, which is the primarily responsible hormone for the development of the udder in mammals. This influence of the number of goats on production is independent of other factors such as the number of lactations, lactation length, age, body weight, among others.

Additionally, Byatt et al. [23] studied the positive effect of placental lactogen by stimulating milk production in cattle using increasing doses of recombinant placental lactogen. In turn, authors such as Delouis et al. [24], Salvador and Martínez [21], or Martí Vicent [22] identified the pivotal role of lactogen but also suggested it may not be the only hormone responsible for the development of the mammary gland during pregnancy. In these regards, other hormones such as prolactin and fetus-placental and ovarian steroids and other hormones involved in general metabolism may be involved as well.

The number of kids did not affect fat and protein percentages in Murciano-Granadina goats’ milk in our study, which agrees with the results by other authors such as Olechnowicz and Sobek [25], Ibnelbachyr et al. [26] and others [21,27]. Oppositely, some authors such as Gómez et al. [28], Ciappesoni et al. [29] and Bagnicka et al. [30] have reported significant differences across the different possibilities within the number of kids factor. For instance, fat and protein levels have been reported to be lower in goats with a higher number of kids as an indirect effect of the increase in milk production. Still, the results of different studies are influenced by different factors, for example, feeding, body condition; hence, it is not easy to distinguish which influences may indeed have a genetic origin from those which may not.

Although component percentages were not significantly different as the number of kids increased, significantly higher means were found for total protein and fat production. These trends were also described for multiple births, Florida goats [31], when a predominant number of female kids was present in litters. As a result, females kidding multiple female kids presented not only higher milk yields, which was also supported by our results, but also higher fat, protein and lactose production. Parity season is one of the most commonly reported conditioning factors to influence milk production and composition. This may be due to the different climatic factors which can affect the animals during each season. The effect of the different factors may become accentuated in animals, which present a marked seasonality, for which temperature, photoperiod, humidity may have a greater impact. Many of these variations are related to pasture curve and food availability during each season, which will influence not only milk production but also lactation length and the shapes which lactation curve describes [32]. Pizarro et al. [33] found similar results to those in the present study on Murciano-Granadina goats and ascribed the lack of significance in literature found for protein and fat contents to studies only recording the information from animals with single and twin-parities, hence the influence of higher numbers of kids may be disregarded and responsible for our findings.

Summer lactations were more productive than autumn lactations, and these, in turn, were better than those of winter, with the worst lactations occurring after the births in spring. This is because late summer births often result in high and longer-lasting initial productions. Furthermore, as seen in Figure 4, even if this occurs, lactation peaks do not reach higher levels above the initial productions and are reached within a few days after the lactation started.

Likewise, the slope of the curve, and therefore the decline of production, is not so sharp. On the contrary, births in spring are followed by lactations characterized by higher peaks, which is due to the positive effect of the abundance of food at this time, which coincides with the higher demands of the animals. Nevertheless, the descending phase of the curve is sharper since it is simultaneous to the summer. During summer months, the quality and quantity of pastures are the lowest of the year, which will also determine these lactations may present the shortest average lactation lengths. Lactations after the births occurring in winter and autumn have intermediate characteristics. These results are very similar to those obtained by Deroide et al. [34]; Pérez [12]; León et al. [10]; Fernández et al. [35]; Verdejo et al. [36] and Carrizosa et al. [37] in Murciano-Granadina goats and Sánchez et al. [38] in Florida goats.

In terms of protein, the highest percentage was obtained in summer, followed by autumn, with the lowest percentages being reported for winter and spring. These results are similar to those obtained by other authors, such as those published by Carrizosa et al. [37], Gómez et al. [28], Fernández et al. [35], Pérez [12] and Deroide et al. [34] with Murciano-Granadina goats. For these authors, the lowest percentages were found for parities occurring during the spring months, with the highest percentages being reported after the parities taking place around the month of September.

The highest fat percentages are reached in summer and autumn, with the lowest ones being reported for winter and spring. Similar results were found in the studies by Gómez et al. [28] and Deroide et al. [34] in which the highest fat percentages were found for births taking place between August and December, while the lowest were reported for the months between January and July.

Such fluctuations are in line with the results by other authors such as Martínez Navalón and Peris Ribera [11] who reported similar fluctuations in average milk productions during the period 1995–2001, also reported by CAPRIGRAN and ACRIMUR associations between 2002 and 2012. Similar results, such as those obtained by Lôbo et al. [39], also reported oscillations in average milk productions over the years that their study took place.

In this context, milk production and average daily production describe fluctuations. In recent years, milk production peaks close to 350 kg have been reached, and then declined. This has translated into maximum production differences of around 80 kg. On the other hand, the fat percentage described an upward trend from 1999 on, reporting an average of 5.35%. For protein percentage, average values of around 3.68% were found in recent years.

Protein percentages also presented an upward trend. However, oscillations were reported for the period of the study. Other authors, such as Byatt et al. [23], found significant differences per year of lactation for fat and protein percentage in milk. Avilés et al. [40] and Martínez Navalón and Peris Ribera [11] also reported an upward trend for fat and protein percentages, both in primiparous and multiparous goats. Likewise, Lôbo et al. [39] reported fluctuations in fat and protein percentages, describing upward trends and reporting the highest percentages found in the literature for the last years comprised in the study. This finding may be ascribed to the effects of the selection of individuals by breeders based on dams’ fat and protein indices.

The significant effect of livestock on the variables considered in the present study may be a sign of the underlying variability. Variability found may be ascribed to the large difference of conditions across populations, among other factors, such as operating system, reproductive, food and/or health management or infrastructure. Even microclimatic conditions to which each population is exposed [21] may be responsible for a certain degree of the variability that can be observed. Among the studies in literature, the best representation of interfarm variability was reported by Pérez [12], who concluded that 20% of interfarm variations among the herds comprising CAPRIGRAN and ACRIMUR associations were due to the effect of livestock. Authors such as Palma [41] found variability may depend on the conditions to which each livestock unit is exposed or to the ones which are present where herds are located. In this context, food management may be one of the pivotal factors to explain the greatest interherd variability. Simultaneously, according to Kučević et al. [42] and Sandrucci et al. [43], these significant differences may also occur in milk composition when different herds are compared.

## 5. Conclusions

The compositional quality of the milk of Murciano-Granadina goats stands out among the Spanish and International dairy goat panorama. Contextually, even if lower milk yields are reported, higher total contents of protein and fat are present, which may translate into an increased technological value of this milk for the production of cheese and other milk derivatives. Highly prolific goats, responsible for this increased component production, may result from the application of adequate reproductive management seeking the maximization of selection practices. The effect of the different factors may become accentuated in animals, which present a marked seasonality, for which temperature, photoperiod, humidity may have a greater impact. Analyzing the variability of the different parameters studied over the 20 years of study, we found significant differences in both milk production and its composition. This effect may be related to climatic differences, which, as aforementioned, has often been studied through the assessment of the effect of seasons on animals. It may also be due to the progressive incorporation of farms with a lower level of production into the official dairy control.

## Figures and Tables

**Figure 1 animals-10-02115-f001:**
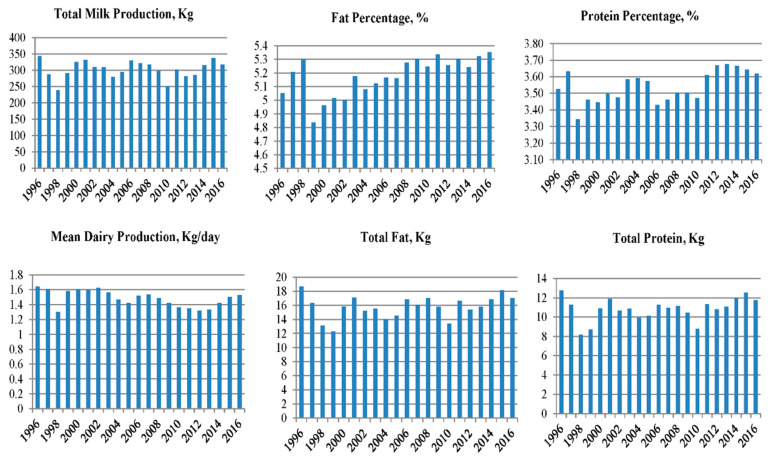
Evolution of the variables studied across parity years.

**Figure 2 animals-10-02115-f002:**
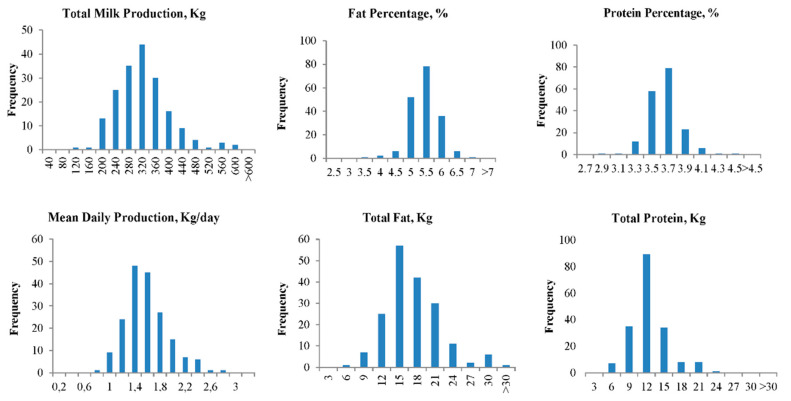
Frequency histograms for the variables studied across farms.

**Figure 3 animals-10-02115-f003:**
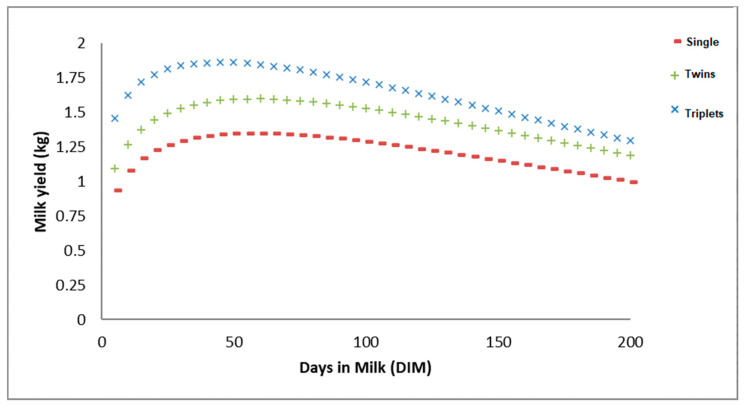
Linearized Wood’s function lactation curves across the number of kids levels.

**Figure 4 animals-10-02115-f004:**
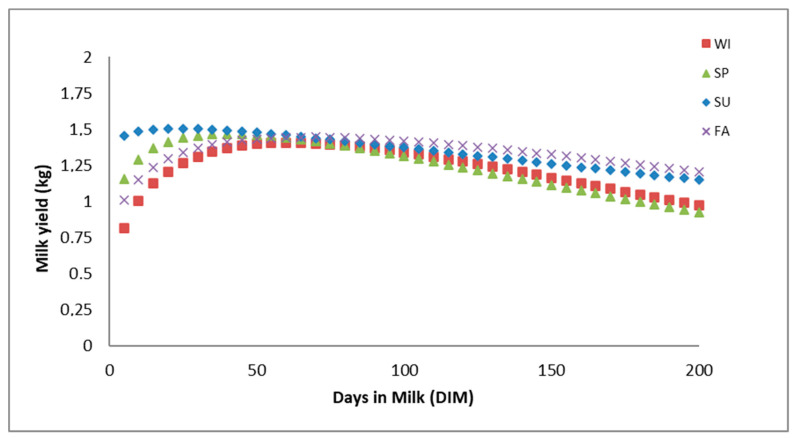
Linearized Wood’s function lactation curves across kidding season levels (WI: winter, SP: spring; SU: summer; FA: autumn/fall).

**Table 1 animals-10-02115-t001:** Data exclusion criteria.

Criteria		Number of Controls	Number of Lactations
Initial validation	Initial number	2,834,425	
	Control type (A4)		
	Number of kids ≤ 5		
	Parturition years 1996–2016		
Control validation	≥3 controls/lactation	411,972	
	First control < 70 days		
	Interval between controls < 70 days Last control < 300 days		
Lactation	First lactation	137,927	22,932

**Table 2 animals-10-02115-t002:** Number of lactations per number of kids, year and season.

Fixed Effects (Factors)	Number of Levels	Frequency
Number of kids	Single	15,481
	Twin	7157
	Triplets	259
Parity year	1996	309
	1997	521
	1998	280
	1999	402
	2000	768
	2001	871
	2002	1255
	2003	1001
	2004	927
	2005	651
	2006	1535
	2007	2824
	2008	1745
	2009	2113
	2010	1699
	2011	1580
	2012	1515
	2013	1129
	2014	805
	2015	598
	2016	369
Parity season	Winter	9417
	Spring	3745
	Summer	2861
	Autumn	6874

**Table 3 animals-10-02115-t003:** Significance level (*p*-value) for the farm, parity year and season and number of kid factors, determination coefficient (R^2^) and residual standard deviation (RSD) for the variables considered in the study.

	Farm	Parity Year	Number of Kids	Parity Season	R^2^	RSD
Total Milk Production, kg	<0.0001	<0.0001	<0.0001	<0.0001	31.40	125.740
Mean daily milk production, kg/day	<0.0001	<0.0001	<0.0001	<0.0001	33.10	0.456
Mean lactation length, days	<0.0001	<0.0001	<0.0001	<0.0001	27.00	24.650
Total Protein Production, kg	<0.0001	<0.0001	<0.0001	<0.0001	33.91	4.454
Total Fat Production, kg	<0.0001	<0.0001	<0.0001	<0.0001	33.56	6.400
Protein, %	<0.0001	<0.0001	0.2912	<0.0001	27.75	0.300
Fat, %	<0.0001	<0.0001	0.8533	<0.0001	23.31	0.700

**Table 4 animals-10-02115-t004:** Least square means values for studied variables across the different levels of the number of kids factor.

Variables		Number of Kids		Overall, Mean
	Single	Twin	Triplets	
Total milk production, kg	265.1 ^c^ ± 2.7	306.1 ^b^ ± 2.9	338.9 ^a^ ± 8.4	303.37 ± 4.67
Mean dairy milk production, kg/day	1.34 ^c^ ± 0.01	1.51 ^b^ ± 0.01	1.62 ^a^ ± 0.03	1.49 ± 0.02
Mean lactation length, days	195.9 ^c^ ± 1.1	199.8 ^b^ ± 1.2	206.5 ^a^ ± 3.4	200.73 ± 1.9
Total protein production, kg	9.43 ^c^ ± 0.1	10.85 ^b^ ± 0.1	12.20 ^a^ ± 0.3	10.83 ± 0.17
Total fat production, kg	13.64 ^c^ ± 0.2	15.79 ^b^ ± 0.2	17.96 ^a^ ± 0.5	15.8 ± 0.3
Protein, %	3.56 ^a^ ± 0.01	3.55 ^a^ ± 0.01	3.53 ^a^ ± 0.02	3.55 ± 0.01
Fat, %	5.17 ^a^ ± 0.02	5.17 ^a^ ± 0.02	5.19 ^a^ ± 0.05	5.18 ± 0.03

^a–c^ The same superindex letter is indicative of the lack of significant differences in the least-square means for the levels sharing the same initial.

**Table 5 animals-10-02115-t005:** Least square means for studied variables across the different levels of parity season factor.

Variables	Kidding Season			Overall, Mean	
	Winter	Spring	Summer	Autumn	
Total milk production, kg	287.2 ^c^ ± 3.8	278.5 ^d^ ± 4.2	329.9 ^a^ ± 4.4	318 ^b^ ± 3.9	299.35 ± 3.16
Mean dairy milk production, kg/day	1.48 ^b^ ± 0.01	1.45 ^c^ ± 0.02	1.50 ^a^ ± 0.02	1.52 ^a^ ± 0.01	1.59 ± 0.02
Mean lactation length, day	192.4 ^c^ ± 1.6	188.7 ^d^ ± 1.7	214.4 ^a^ ± 1.8	207.2 ^b^ ± 1.6	197.51 ± 1.27
Total protein production, kg	10.04 ^c^ ± 0.2	9.74 ^d^ ± 0.2	12.08 ^a^ ± 0.2	11.45 ^b^ ± 0.2	10.89 ± 1.1
Total fat production, kg	14.38 ^c^ ± 0.2	14.08 ^c^ ± 0.2	17.96 ^a^ ± 0.3	16.75 ^b^ ± 0.2	15.9 ± 0.19
Protein, %	3.45 ^c^ ± 0.01	3.45 ^c^ ± 0.01	3.66 ^a^ ± 0.01	3.62 ^b^ ± 0.01	3.62 ± 0.41
Fat, %	4.97 ^c^ ± 0.02	4.98 ^c^ ± 0.03	5.41 ^a^ ± 0.03	5.35 ^b^ ± 0.03	5.23 ± 0.07

^a–d^ The same superindex letter is indicative of the lack of significant differences in the least-square means for the levels sharing the same initial.

**Table 6 animals-10-02115-t006:** Average values across the number of kids levels for the parameters of the lactation curve (a, b and c); peak day (PD); peak production (PP); persistence percentage at 100 (P%100), 150 (P%150) and 200 days (P%200).

Parameters	Number of Kids		
	Single	Twin	Triplets
a	0.650	0.749	1.104
b	0.240	0.247	0.186
c	0.00421	0.00425	0.00414
PD	57.0	58.2	45.0
PP	1.351	1.598	1.862
P%100	95.695	95.981	92.509
P%150	89.305	89.388	87.681
P%200	86.814	86.820	85.778

**Table 7 animals-10-02115-t007:** Average values across kidding season levels for the parameters of the lactation curve (a, b and c); peak day (PD); peak production (PP); persistence percentage at 100 (P%100), 150 (P%150) and 200 days (P%200).

Parameters	Kidding Season			
	Winter	Spring	Summer	Autumn
a	0.483	0.862	1.358	0.729
b	0.343	0.198	0.048	0.213
c	0.00559	0.00488	0.00211	0.00312
PD	61.410	40.540	22.897	68.222
PP	1.407	1.470	1.506	1.446
P%100	95.931	89.874	93.054	99.158
P%150	86.914	84.907	91.769	93.274
P%200	83.471	82.953	91.249	90.967

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
