# Peer review of "Conditioning Factors of Linearized Wood’s Function Lactation Curve Shape Parameters, Milk Yield, Fat and Protein Content in Murciano-Granadina Primiparous Does"

_animals, 2020, doi:10.3390/ani10112115_

Round 1

Reviewer 1 Report

The paper from Bermejo et al deals with the effect of some environmental factors on milk traits of a Spanish dairy goat.

The paper is quite interesting but to be honest I made a tremendous effort in going through the manuscript

The English really needs an extended revision. I strongly suggest you to use shorter statements . You can do this by avoiding using “and”. You can split very long statements

I have re-phrased line 43-48 just to give you an example.  

Line 43-48. The Murciano-Granadina Goat is the most important Spanish dairy goat, both in the number of individuals and production level. There are more than 500,000 animals, of which 104,010 registered to the official herd book. They are spread across Spain, especially in Andalusia, Murcia and Castilla la Mancha, but an increasing number of animals are now reared abroad, mainly in other European countries, North Africa, Latin America and the United States [1].

Another suggestion I would like to give you is to use a little bit the compound nouns. There are a lot of examples in your manuscript. For instance:

Line 55: the average production of milk = the average milk production ; duration of lactation = lactation lengh

Line 63: capacity for adaptation = adaptive capacity

Line 65: in the prevention of fires = in fire prevention

Please revise it carefully

Line 54 : births: I think that parity is more appropriate

Line 57-59. Please rephrase

Line 72-73. ..detect unexpected deviations at herd level as consequences of a …

Line 81: please provide some references

Line 85: controls = different milk recording schemes

Line 88: delete ‘in literature’

Line 89: delete “TO this aim”

Line 105. After editing a total of …. was available

Line 107: deliveries? Please find a more appropriate term and try to rephrase. Anyway, I would comment table 2 in the Results section

Line 114 milk production & C are not factors but traits. I’d rather use Traits and fixed effects (in table 2 as well)

Line 123-124. Please provide citation for the methodology. ‘thanks to” = following / according to

Line 129: IN which= where

Line 134-135 = ? not clear. Please re-phrase untile Line 141

Line 147-175: I suggest to join 2.5 and 2.6 into a subsection like Statistical Analyses.  I strongly suggest to double check statement’s structure. SAS needs a reference

Line 181 I would include comments on table 2 and then move on to table 3.

Line 182: Results from analyses of variance for all traits considered are in table 3. (Please do not include the Mean). Y

Line 188: Do you mean that you check normality ? IF it is, I think this is M & M. Please be consistent: use traits or variables across all the manuscript

Line 194 Type of kidding= Number of Kids. Do you mean Least Square Means? A note on what letters mean is mandatory

Line 198. Likewise ???

Line 195 – 209. Please re-phrase according to previous suggestions (i.e. short statements)

Line 251-236. Please re-phrase according to previous suggestions (i.e. short statements). Take care of the grammar. Some statements are really hard to follow because of the grammar. An example “This gives us quite similar results in winter and spring in terms of fat and protein, being in these two seasons where the lowest quantities were reached, and higher production and higher proportions in summer and autumn, reaching the highest yields in the summer” this very last statement is really hard to understand. 

Line 260: please define “a”

Line 263: …’the highest productions are obtained in the peak’ ?  What do you mean exactly? Once again the statement is too long.

Line 307= first birth? Primiparous

Line 312-316 Our results agree well with…

Line 442 : multiple births? Do you mean number of kids?

Author Response

Reviewer 1

Comments and Suggestions for Authors

The paper from Bermejo et al deals with the effect of some environmental factors on milk traits of a Spanish dairy goat.

The paper is quite interesting but to be honest I made a tremendous effort in going through the manuscript

The English really needs an extended revision. I strongly suggest you to use shorter statements . You can do this by avoiding using “and”. You can split very long statements

I have re-phrased line 43-48 just to give you an example.  

Line 43-48. The Murciano-Granadina Goat is the most important Spanish dairy goat, both in the number of individuals and production level. There are more than 500,000 animals, of which 104,010 registered to the official herd book. They are spread across Spain, especially in Andalusia, Murcia and Castilla la Mancha, but an increasing number of animals are now reared abroad, mainly in other European countries, North Africa, Latin America and the United States [1].

Response: Reviewer suggestion was applied.

Another suggestion I would like to give you is to use a little bit the compound nouns. There are a lot of examples in your manuscript. For instance:

Response: As suggested by the reviewer compound names were used and exchanged when necessary.

Line 55: the average production of milk = the average milk production ; duration of lactation = lactation lengh

Line 63: capacity for adaptation = adaptive capacity

Line 65: in the prevention of fires = in fire prevention

Response: As suggested by the reviewer compound names were used and exchanged when necessary.

Please revise it carefully

Line 54 : births: I think that parity is more appropriate

Response: The word parity was used across body text and Tables.

Line 57-59. Please rephrase

Response: rephrased.

Line 72-73. ..detect unexpected deviations at herd level as consequences of a …

Response: Changed.

Line 81: please provide some references

Response: Added.

Line 85: controls = different milk recording schemes

Response: Changed.

Line 88: delete ‘in literature’

Response: Deleted.

Line 89: delete “TO this aim”

Response: Deleted

Line 105. After editing a total of …. was available

Response: Changed

Line 107: deliveries? Please find a more appropriate term and try to rephrase. Anyway, I would comment table 2 in the Results section

Response: parity was used instead of deliveries.

Line 114 milk production & C are not factors but traits. I’d rather use Traits and fixed effects (in table 2 as well)

Response: Changed

Line 123-124. Please provide citation for the methodology. ‘thanks to” = following / according to

Response: Citation added.

Line 129: IN which= where

Response: Changed.

Line 134-135 = ? not clear. Please re-phrase untile Line 141

Response: rephrased.

Line 147-175: I suggest to join 2.5 and 2.6 into a subsection like Statistical Analyses.  I strongly suggest to double check statement’s structure. SAS needs a reference

Response: Reviewer suggestions were followed and both sections were conjoined into the statistical analyses section.

Line 181 I would include comments on table 2 and then move on to table 3.

Response: Table 2 reports the outputs of the process of appliation exclussion criterio to obtain the study sample and is mentioned in the text in M&M. We think Table 2 and 3 are mentioned in the correct order.

Line 182: Results from analyses of variance for all traits considered are in table 3. (Please do not include the Mean).

Response: Mean was removed from Table 3.

Line 188: Do you mean that you check normality ? IF it is, I think this is M & M. Please be consistent: use traits or variables across all the manuscript

Response: We agree, we moved it. The use of variables and traits was made consistent across the manuscript.

Line 194 Type of kidding= Number of Kids. Do you mean Least Square Means? A note on what letters mean is mandatory

REsposne: type of kidding was changed to number of kids. We chaged mean to Least Square Means and a note was added.

Line 198. Likewise ???

Resposne: Deleted.

Line 195 – 209. Please re-phrase according to previous suggestions (i.e. short statements)

Response: Rephrased.

Line 251-236. Please re-phrase according to previous suggestions (i.e. short statements). Take care of the grammar. Some statements are really hard to follow because of the grammar. An example “This gives us quite similar results in winter and spring in terms of fat and protein, being in these two seasons where the lowest quantities were reached, and higher production and higher proportions in summer and autumn, reaching the highest yields in the summer” this very last statement is really hard to understand. 

Response: Rephrased. We rechecked and corrected the mnauscript seeking for grmmar mistakes and typos and to improve readability.

Line 260: please define “a”

Response: Defined.

Line 263: …’the highest productions are obtained in the peak’ ?  What do you mean exactly? Once again the statement is too long.

Response: Statement was rewritten.

Line 307= first birth? Primiparous

Response: Changed.

Line 312-316 Our results agree well with…

Response: Changed.

Line 442 : multiple births? Do you mean number of kids?

Response: Yes, changed.

Reviewer 2 Report

The aim of the reviewed work  was to evaluate the performance of the Wood model to describe the characteristics of lactation curves of Murciano-Granadina goats. The study of milk production can be performed using mathematical functions that estimate the level of production reached over time. By fitting these mathematical functions, the data obtained from the controls can be used to predict milk yield and trace the evolution of total production. Among the mathematical models that have been commonly used to fit lactation curves, the equation proposed by Wood in 1967 is one of the most frequently used in literature . A total of 137,927 controls of 22,932 Murciano-Granadina first lactation goats (measured between 1996-2016) were evaluated to determine the influence of kidding type, season, year and farm on total milk yield, daily milk yield, lactation length, total production of fat and protein, and percentages of fat and protein. Production control was carried out monthly using the A4 method, following the recommendations of the International Committee for the Control of Animal Performance (ICAR). All factors analyzed had a significant effect on the variables studied, except for the influence of kidding type on the percentages of fat and protein, where the variation was very small. The results obtained are very different from the others and have not been sufficiently documented by the authors. Especially since there were significant differences for the influence of kidding type on the total protein production and total fat production. Overall mean values ​​for each feature can be inserted in Tables 4 and 5 for better interpretation of results. Figure 2 is identical to Figure 1, which is probably wrong. No table or drawing illustration for the effect of the year of kidding and the effect of farm. Figure 8 referred to in the text is not attached.The conclusions are poorly documented and are more suitable for discussing the results.

Author Response

Reviewer 2

The aim of the reviewed work was to evaluate the performance of the Wood model to describe the characteristics of lactation curves of Murciano-Granadina goats. The study of milk production can be performed using mathematical functions that estimate the level of production reached over time. By fitting these mathematical functions, the data obtained from the controls can be used to predict milk yield and trace the evolution of total production. Among the mathematical models that have been commonly used to fit lactation curves, the equation proposed by Wood in 1967 is one of the most frequently used in literature. A total of 137,927 controls of 22,932 Murciano-Granadina first lactation goats (measured between 1996-2016) were evaluated to determine the influence of kidding type, season, year and farm on total milk yield, daily milk yield, lactation length, total production of fat and protein, and percentages of fat and protein. Production control was carried out monthly using the A4 method, following the recommendations of the International Committee for the Control of Animal Performance (ICAR). All factors analyzed had a significant effect on the variables studied, except for the influence of kidding type on the percentages of fat and protein, where the variation was very small.

The results obtained are very different from the others and have not been sufficiently documented by the authors. Especially since there were significant differences for the influence of kidding type on the total protein production and total fat production.

Response: Further discussion as proposed by the reviewer was added, specially focusing on the finding of significant differences for total protein and fat contents.

Overall mean values ​​for each feature can be inserted in Tables 4 and 5 for better interpretation of results.

Response: overall mean values were added in both tables.

Figure 2 is identical to Figure 1, which is probably wrong.

Response: We agree with reviewer, we added the correct Figure 2.

No table or drawing illustration for the effect of the year of kidding and the effect of farm.

Response: Figures 1 and 2 were added to follow the reviewer’s suggestion. Then remaining Tables were numbered accordingly.

Figure 8 referred to in the text is not attached.

Response: This was a typo and was removed.

The conclusions are poorly documented and are more suitable for discussing the results.

Response: Conclusions were rewritten.